# Differential Effects of High Fat Diets on Resilience to H_2_O_2_-Induced Cell Death in Mouse Cerebral Arteries: Role for Processed Carbohydrates

**DOI:** 10.3390/antiox12071433

**Published:** 2023-07-16

**Authors:** Charles E. Norton, Rebecca L. Shaw, Steven S. Segal

**Affiliations:** 1Department of Medical Pharmacology and Physiology, University of Missouri, Columbia, MO 65212, USAsegalss@health.missouri.edu (S.S.S.); 2Dalton Cardiovascular Research Center, Columbia, MO 65211, USA; 3Department of Biomedical Sciences, University of Missouri, Columbia, MO 65201, USA; 4Department of Biomedical, Biological and Chemical Engineering, University of Missouri, Columbia, MO 65211, USA; 5Department of Nutrition and Exercise Physiology, University of Missouri, Columbia, MO 65211, USA

**Keywords:** smooth muscle cells, endothelial cells, mitochondrial membrane potential, Src family kinases, transient receptor potential (TRP) channels

## Abstract

High fat, western-style diets increase vascular oxidative stress. We hypothesized that smooth muscle cells and endothelial cells adapt during the consumption of high fat diets to become more resilient to acute oxidative stress. Male C57Bl/6J mice were fed a western-style diet high in fat and processed carbohydrates (WD), a high fat diet that induces obesity (DIO), or their respective control (CD) and standard (SD) diets for 16 weeks. Posterior cerebral arteries (PCAs) were isolated and pressurized for study. During acute exposure to H_2_O_2_ (200 µM), smooth muscle cell and endothelial cell death were reduced in PCAs from WD, but not DIO mice. WD selectively attenuated mitochondrial membrane potential depolarization and vessel wall Ca^2+^ influx during H_2_O_2_ exposure. Selective inhibition of transient receptor potential (TRP) V4 or TRPC3 channels reduced smooth muscle cell and endothelial cell death in concert with the vessel wall [Ca^2+^]_i_ response to H_2_O_2_ for PCAs from CD mice and eliminated differences between CD and WD. Inhibiting Src kinases reduced smooth muscle cell death along with [Ca^2+^]_i_ response to H_2_O_2_ only in PCAs from CD mice and eliminated differences between diets. However, Src kinase inhibition did not alter endothelial cell death. These findings indicate that consuming a WD, but not high fat alone, leads to adaptations that limit Ca^2+^ influx and vascular cell death during exposure to acute oxidative stress.

## 1. Introduction

Western-style diets (WD) are high in fat and processed carbohydrates, which promote obesity [1]. In turn, obesity leads to excessive levels of reactive oxygen species (ROS), thereby inducing oxidative stress in humans and animals [2,3]. Furthermore, consuming processed carbohydrates can augment oxidative stress in obesity [4]. Oxidative stress is associated with cerebral deficits such as memory and behavioral impairments [5], which are linked to apoptosis of neuronal and vascular cells [6]. The consumption of a high fat diet is also a key risk factor for stroke [7,8]. Acute oxidative stress and apoptosis are consequences of ischemic stroke where, upon reperfusion, ROS damage neurons [9] and vascular [10] cells in the brain. Therefore, to limit damage to the cerebral vasculature and the parenchyma it supplies, greater understanding is needed with respect to how the consumption of high fat diets affects the susceptibility of smooth muscle cells (SMCs) and endothelial cells (ECs) to apoptosis when exposed to acute oxidative stress.

H_2_O_2_ is common to multiple pathways of ROS production [11] and elicits apoptosis via the intrinsic pathway [12], which is triggered by an overload of intracellular Ca^2+^ concentration ([Ca^2+^]_i_) leading to increases in mitochondrial Ca^2+^ content and depolarization of mitochondrial membrane potential (ΔΨ_m_) [13]. Loss of ΔΨ_m_ facilitates release of cytochrome C into the cytosol, where it interacts with the apoptosis-activating factor and caspase 9, thereby activating caspase 3, the death protease mediating cell death. In arteries supplying the brain, acute H_2_O_2_ exposure promotes Ca^2+^ entry through transient receptor potential (TRP) channels, which are critical for eliciting cell death through intrinsic apoptosis [14]. As shown in arteries of skeletal muscle, consuming a WD leads to cellular adaptations that reduce Ca^2+^ through TRP channels and thereby attenuates cell death [15]. Whether cerebral arteries adapt to consuming a WD in a similar manner is unknown.

In the present study, we tested whether SMCs and ECs of posterior cerebral arteries (PCAs) from mice consuming a diet high in fat and processed carbohydrate would develop resilience to acute oxidative stress from H_2_O_2_ by evaluating cell death, [Ca^2+^]_i_ and ΔΨ_m_ responses to H_2_O_2_. To determine whether processed carbohydrate was an integral dietary component to developing resilience, a high fat diet that induces obesity (DIO) [16] but is low in processed carbohydrate was evaluated for reference. Complementary experiments tested whether TRP channels mediate the protection of ECs and SMCs of PCAs during H_2_O_2_ exposure.

## 2. Materials and Methods

### 2.1. Animal Care and Use

Experimental procedures were reviewed and approved by the University of Missouri Animal Care and Use Committee (Protocol #17720). Male mice were used for all experiments because vessels from females are more resilient to oxidative stress [15]. Mice were housed on a 12:12 h light–dark cycle at ∼23 °C with fresh water and food available ad libitum. Mice were anaesthetized with ketamine and xylazine (100 kg^−1^ and 10 mg·kg^−1^, respectively; intraperitoneal injection) and killed by decapitation.

### 2.2. Diet Compositions

Male C57Bl/6J mice (4 wk old; purchased from Jackson Laboratories, Bar Harbor, ME, USA) were housed in the University of Missouri animal facility and fed a WD high in fat and processed carbohydrates (calories: 46% fat, 35% carbohydrate (17.5% high-fructose corn syrup, 13% starch, 5% mixed sugars), 19% protein; TestDiet 58Y1 modified with added corn syrup, Richmond, IN, USA) or a control diet (CD; calories: 17% fat, 56% carbohydrate (39% starch, 17% mixed sugars), 27% protein; Formulab Diet 5008, LabDiet, St. Louis, MO, USA) for 16 weeks prior to study [15,17,18].

In complementary experiments, male C57Bl/6J DIO mice (Strain #380050) and their standard diet controls (SD; Strain #380056) were purchased from Jackson Laboratories when ~20 wk old after 16 weeks of being fed their respective diets: DIO mice consumed a high fat diet that was lower in processed carbohydrates (calories: 60% fat, 20% carbohydrate (4% starch, 16% mixed sugars), 20% protein; Research Diets D12492, Brunswick, NJ, USA) and SD mice fed a low fat diet (calories: 10% fat, 70% carbohydrate (31% starch, 39% mixed sugars), 20% protein; Research Diets D12450B) [16]. All mice were studied at ~22 weeks of age.

### 2.3. Preparation of Isolated Posterior Cerebral Arteries

Intact brains were removed from the skull and placed in chilled (4 °C) physiological salt solution (PSS, pH 7.4; containing (in mM): 140 NaCl (Thermo Fisher Scientific; Waltham, MA, USA), 5 KCl (Thermo Fisher), 1 MgCl_2_ (Sigma-Aldrich, St. Louis, MO, USA), 10 HEPES (Sigma), 2 mM CaCl_2_ (Fisher) and 10 glucose (Thermo Fisher)) and pinned onto silicon elastomer (Sylgard 184^®^; Dow Corning, Midland, MI, USA). An unbranched segment (∼2 mm long) of the PCA was dissected from surrounding parenchyma while viewing through a stereomicroscope. Individual PCAs were cannulated onto micropipettes (heat-polished; outer diameter, ~80 µm) and tied in place with a strand of 7–0 silk suture. Once cannulated, arteries were positioned in a tissue chamber (RC-27N; Warner Instrument; Hamden, CT, USA) and superfused at 3 mL min^−1^ with control PSS. Vessels were pressurized to 90 cm H_2_O (∼65 mmHg) and maintained at 37 °C [14].

### 2.4. Vascular ROS Production

To evaluate ROS production within the vessel wall, intact pressurized PCAs were loaded with 5-(and-6-)-chloromethyl-2,7-dichlorodihydro-fluorescein diacetate acetyl ester (DCFH; Cat. #C6827, Fisher Scientific) [15,19]. DCFH was diluted to 15 µM in PSS (final DMSO = 0.5%). The PCA was equilibrated in this solution for 30 min, then superfusion with PSS was restored. Fluorescence images (each 35 ms) were acquired onto a personal computer for 30 min at 5 min intervals using a MV PLAPO 2X objective (NA = 0.5, Olympus, Tokyo, Japan) coupled to a megapixel CCD camera (XR/Mega10, Stanford Photonics, Palo Alto, CA, USA) on an Olympus MVX10 microscope (final magnification, ∼120×). An X-Cite illuminator (model no. 120, Excelitas Technologies, Waltham, MA, USA) provided excitation at 472/30 nm with emission at 525/35 nm. This fluorescent indicator of vascular ROS production has been validated with both positive and negative controls [15].

### 2.5. Cell Death

Prior to cannulation, pipettes were filled with PSS containing the membrane-permeant nuclear dye Hoechst 33,342 (1 µM; Cat. #H1399, Fisher) to identify all cell nuclei and propidium iodide (2 µM; Cat. #4170, Sigma), which permeates membranes of dead and dying cells; respective dyes were thereby introduced into the vessel lumen [12,15]. Time controls have verified that mouse PCAs studied under these conditions exhibit <1% cell death after 50 min when not exposed to H_2_O_2_ [14]. Pressurized PCAs were equilibrated for 20 min in PSS containing vehicle alone or with a pharmacological agent, then exposed to 200 µM H_2_O_2_ (Cat. #H1009, Sigma) for 50 min. Following H_2_O_2_ exposure, superfusion with fresh PSS resumed while the PSS containing the nuclear dyes perfused the lumen (0.1 mL min^−1^, 10 min). Luminal perfusion was halted during H_2_O_2_ exposure because luminal flow reduces cell death and the SMC monolayer does not restrict EC access to H_2_O_2_ delivered from the bath [12].

Cell death was quantified as described [12,14,15]. Fluorescent images of Hoechst 33,342 (blue) and PI (red) were acquired with a 40× water immersion objective (numerical aperture (NA) = 0.80) using appropriate filters and coupled to a DS-Qi2 camera on an E800 microscope using Elements software (version 4.51; all from Nikon). Z-stack images were obtained through the upper half of a vessel segment. EC nuclei were identified by having an oval shape oriented parallel to the vessel axis while SMC nuclei are and thin and orientated perpendicular to the vessel axis.

### 2.6. Mitochondrial Membrane Potential

Pressurized PCAs were loaded from the bath with the mitochondrial-targeted ΔΨ_m_ fluorescent indicator tetramethylrhodamine methyl ester (10 nM in PSS; TMRM, Cat. #T668, Fisher) for 30 min preceding H_2_O_2_ exposure [20,21] and throughout the protocol. TMRM accumulates in the mitochondrial matrix due to the electronegative potential within these organelles; thus, the intensity of fluorescence decreases with depolarization of ΔΨ_m_ [22]. Fluorescence images were acquired as described in Section 2.4 at 1 min intervals for 30 min with excitation at 543/22 nm and emission at 592/40 nm. The protonophore carbonyl cyanide 4-(trifluoromethoxy)phenylhydrazone (FCCP, 10 µM; Cat. #C6827, Sigma) was used as a positive control to verify changes in ΔΨ_m_ [23].

### 2.7. Ca^2+^ Photometry

A pressurized PCA was positioned on an inverted microscope (Eclipse TS100, Nikon) and incubated in a static solution of Fura 2-AM dye (diluted to 1 µM in PSS (final [DMSO] = 0.5%); Cat. #F14158, Fisher) for 40 min. Under these conditions, the dye is primarily incorporated into SMCs. Superfusion with PSS was then resumed for 20 min to wash out excess dye. Using a Nikon Fluor 20× objective (NA = 0.45), the vessel was excited at 340 and 380 nm with emission acquired at 510 nm using an IonOptix system with IonWizard 6.3 software [12]. After recording baseline fluorescence, 200 µM H_2_O_2_ was added to the superfusion solution. F_340_/F_380_ ratios were recorded at 10 Hz for 30 s at 5 min intervals (to limit photobleaching of Fura 2 dye) during 50 min exposure to H_2_O_2_ and the ensuing 30 min wash with control PSS.

### 2.8. Experimental Interventions

Pharmacological agents were added to PSS to evaluate how respective signaling components affected vascular cell death and [Ca^2+^]_i_ responses to H_2_O_2_. TRP4 channels were inhibited with HC-067047 (1 μM in 0.1% EtOH; Cat. #4100, Tocris) [24] and TRPC3 channels were inhibited with 1-[4-[(2,3,3-trichloro-1-oxo-2-propen-1-yl)amino]phenyl]-5-trifluoromethyl)-1H-pyrazole-4-carboxylicacid, ethyl ester (Pyr3, 1 μM in 0.1% EtOH; Cat. #16888, Cayman) [25]. Src family kinases were inhibited with SU6656 (10 μM in 0.1% EtOH; Cat. #6475, Tocris) [26].

### 2.9. Data Analysis and Statistics

The intensity of DCFH fluorescence was evaluated using Image J software (version 1.52a; National Institutes of Health, Bethesda, MD, USA) in a region of interest (ROI; 80 µm × 300 µm) located in the center of a vessel. After subtracting background fluorescence, values for ROS generation (i.e., fluorescence accumulation in arbitrary units) reflect the change (Δ) from baseline within the ROI over time: Δ = (fluorescence at *x* min − fluorescence at 0 min), where *x* denotes 5 min intervals. The rate of ROS generation (dF/d*t*) was determined using linear regression, where F is fluorescence and *t* is time (min). Live and dead cell nuclei were counted manually using Image J software within a 80 × 300 µm ROI, which contained ~50 ECs and ~50 SMCs [14]. Cell death was calculated as: (# of red nuclei/# of blue nuclei) × 100%. Quantification of ΔΨ_m_ was evaluated within an 80 × 300 µm ROI by assessing TMRM fluorescence relative to the initial baseline fluorescence (F/F_0_). [Ca^2+^]_i_ values within the microscope field of view (~300 × 300 µm) are expressed as the change in F_340_/F_380_ (Δ340/380) from baseline (0 min) at each 5 min interval following subtraction of background fluorescence recorded before dye loading. Student’s *t* tests or ANOVA (Prism 9, GraphPad Software, La Jolla, CA, USA) were used to analyze data as appropriate with Bonferroni’s test for post hoc comparisons. *p <* 0.05 was considered statistically significant. Summary data are displayed as means ± SE, where *n* indicates the number of vessels (each from a separate mouse) in an experimental group.

## 3. Results

### 3.1. Effects of High Fat Diets on Vascular Oxidative Stress

High fat diets result in weight gain, insulin resistance, and vascular oxidative stress [17,18,27]. In the present study, both WD and DIO augmented weight gain (Table 1). The baseline ROS production (DCFH fluorescence accumulation) was greater in PCAs from WD vs. CD mice (Figure 1a,b). There was also a trend (*p* = 0.06) for DIO to elevate ROS production above that of SD mice (Figure 1c,d).

### 3.2. WD, but Not DIO, Increases Resilience to H_2_O_2_

Cell death prior to H_2_O_2_ exposure was minimal for all groups (1–3%). Following H_2_O_2_ exposure (200 µM; 50 min), SMC and EC death were reduced in PCAs from mice fed WD vs. CD mice (Figure 2a–d). In contrast, cell death in PCAs from mice fed DIO was not different from those fed SD (Figure 2e,f), which also had a low incidence of cell death.

Depolarization of ΔΨ_m_ is a key signaling event mediating cell death [28,29]. Changes in ΔΨ_m_ were evaluated with TMRM fluorescence. In PCAs from CD mice, H_2_O_2_ progressively depolarized ΔΨ_m_ (Figure 3) by ~60% over 30 min (Figure 3a). In PCAs from WD mice, depolarization to H_2_O_2_ was reduced to ~30%, illustrating a protective effect of this diet. In contrast, ΔΨ_m_ depolarization to H_2_O_2_ was not different between PCAs from DIO and SD mice (Figure 3b); both were similar to the response of PCAs from CD mice. There were no significant differences in baseline TMRM fluorescence among PCAs from mice consuming respective between diets. In PCAs from CD mice, FCCP was used as a positive control to depolarize ΔΨ_m_ (ΔF/F_0_ = −0.66 ± 0.03, *n* = 3), and in the absence of H_2_O_2_ or FCCP, TMRM fluorescence remained stable for 30 min (ΔF/F_0_ = −0.05 ± 0.02, *n* = 4).

### 3.3. WD Attenuates the [Ca^2+^]_i_ Response Induced by H_2_O_2_

Excessive levels of [Ca^2+^]_i_ contribute to ΔΨ_m_ depolarization and vascular cell death [14,30,31]. For PCAs from CD mice, [Ca^2+^]_i_ increased progressively during H_2_O_2_ exposure and nearly recovered during washout (Figure 4a); the peak [Ca^2+^]_i_ response to H_2_O_2_ was reduced by ~50% in PCA from WD mice. In contrast, the [Ca^2+^]_i_ response to H_2_O_2_ was not different between PCAs from DIO vs. SD mice (Figure 4b). Baseline [Ca^2+^]_i_ was not different between groups and [Ca^2+^]_i_ remains constant throughout the ~90 min protocol in the absence of H_2_O_2_ [14]. Because DIO was not different from SD for cell death (Figure 2e,f), ΔΨ_m_ (Figure 3b), or [Ca^2+^]_i_ (Figure 4b) in response to H_2_O_2_, additional experiments focused on WD vs. CD.

### 3.4. WD Diminishes Ca^2+^ Influx through TRP Channels Induced by H_2_O_2_

TRPC3 and TRPV4 channels are integral to Ca^2+^ entry and cell death in response to H_2_O_2_ exposure [14]. Inhibition of TRPV4 channels with HC-067047 (1 µM) nearly abolished SMC death in PCAs from mice fed CD without affecting SMC death in PCAs from WD mice, which were resilient to H_2_O_2_ (Figure 5a). EC death was similarly attenuated in PCAs from CD mice, but not from WD mice (Figure 5b). TRPV4 channel inhibition reduced Ca^2+^ entry in vessels from CD mice (Figure 5c) and eliminated differences in cell death between dietary groups. This effect of HC-067047 was not observed in PCAs from WD mice, consistent with their attenuated Ca^2+^ entry.

To test the effects of TRPC3 channel inhibition on cellular responses to H_2_O_2_, PCAs were treated with Pyr3 (1 µM), which reduced SMC death (Figure 6a) and EC death (Figure 6b) in vessels from CD, but not WD, mice. Similar to the effects of TRPV4 inhibition (Figure 5), differences in both EC and SMC death in PCAs were eliminated by Pyr3. Inhibition of TRPC3 channels also reduced the [Ca^2+^]_i_ response to H_2_O_2_ in vessels from CD mice (Figure 6c). The increase in [Ca^2+^]_i_ was also attenuated by Pyr3 in vessels from WD mice, albeit to a lesser extent than in vessels from CD mice. Furthermore, Pyr3 eliminated differences in [Ca^2+^]_i_ responses of PCAs to H_2_O_2_ between CD and WD mice (Figure 6c).

### 3.5. Src Kinases Contribute to Cell Death during H_2_O_2_ Exposure

Oxidative stress can activate Src family kinases to enhance TRP channel activity [32]. In PCAs from CD mice, the Src kinase antagonist SU6656 (10 µM) reduced SMC (Figure 7a), but not EC death (Figure 7b) in response to H_2_O_2_. In PCAs from WD mice, SU6656 had no further effect. SU6656 reduced the [Ca^2+^]_i_ response to H_2_O_2_ in PCAs from CD, but not WD, mice (Figure 7c).

## 4. Discussion

We evaluated the resilience of cerebral arteries from 22 wk old male mice fed western-style (WD) and high fat (DIO) diets during 50 min exposure to H_2_O_2_ (200 µM). Our key findings are that following ~16 wk of feeding: (1) WD, but not DIO, enhanced basal ROS production in PCAs compared to the respective control diets; (2) WD, but not DIO, attenuated SMC and EC death; (3) WD, but not DIO, attenuated ΔΨ_m_ depolarization; (4) WD attenuated Ca^2+^ entry through the TRPV4 and TRPC3 channels; and (5) Src kinases contributed to SMC, but not EC, death in PCAs from mice fed the control diet (CD) vs. WD. We propose that cerebral arteries develop resilience to oxidative stress during prolonged consumption of a western-style diet that is high in processed carbohydrates as well as fats. Remarkably, this adaptation preserves vascular cell integrity during acute oxidative stress imposed by H_2_O_2_.

### 4.1. Effects of High Fat Diet on Oxidative Stress and Vascular Cell Death

Oxidative stress and cell death are integral to the pathogenesis of vascular disease, stroke, and traumatic brain injury [10,33,34,35]. Obesity is an independent risk factor that may augment the adverse effects of hypertension, diabetes, and hyperlipidemia on the vasculature [5]. Basal ROS production was greater in PCAs from WD mice vs. CD mice (Figure 1), consistent with the effects of WD in the aorta [36] and skeletal muscle resistance arteries [15]. Although there was a similar trend for PCAs from DIO mice, ROS production was not statistically different from SD mice despite similar increases in BW for WD and DIO mice (Table 1). Finding here that WD augments ROS production in mouse cerebral arteries is consistent with the increased oxidative stress in obese humans consuming processed carbohydrate [4]. Because our experimental design exposes a vessel to constant oxidative stress (200 μM H_2_O_2_ in the superfusion solution), it seems unlikely that differences in antioxidant capacity are responsible for the differences in cell death associated with WD. However, this possibility cannot be excluded. Chronic oxidative stress can upregulate antioxidant defenses including the transcription factor Nrf2 and thereby protect mitochondria [37,38]. To resolve the question of how WD and DIO may differentially modify the antioxidant response in the cerebral vessel wall will require further study, as will identifying the source(s) of ROS production. Whether or not the protection from H_2_O_2_ includes an antioxidant response (or other adaptations), chronic elevation of oxidative stress appears to be integral to greater resilience of SMCs and ECs in the arterial wall during acute exposure to H_2_O_2_.

Vessels from males were studied for the present experiments because those from females are intrinsically protected during H_2_O_2_ exposure [12,15]. Finding that H_2_O_2_ elicited similar levels of death in SMCs and ECs of PCAs (Figure 2) differs from our previous observations that ECs are more resilient than SMCs to H_2_O_2_ [14]. In agreement with reports of chronic oxidative stress promoting vascular resilience [12,15,19], WD (but not DIO) increased SMC and EC survival during H_2_O_2_ exposure, indicating a distinct effect of consuming processed carbohydrates (high fructose corn syrup). Furthermore, depolarization of ΔΨ_m_ was attenuated in PCAs from WD mice, whereas ΔΨ_m_ depolarization to H_2_O_2_ prevailed in DIO mice (Figure 3). In skeletal muscle, high levels of fructose lead to mitochondrial dysfunctions including decreases in DNA content, impaired energy metabolism, and decreased activity of respiratory complexes [39]. These reductions in functional capacity of mitochondria would be maladaptive to vascular cells yet may be beneficial in the context of limiting ΔΨ_m_ depolarization to H_2_O_2_. However, as shown in pancreatic β cells, nutrient excess can augment ΔΨ_m_ [40], which may limit depolarization to H_2_O_2_. Further study is required to identify the specific effects of high fat diets on mitochondrial function in the cerebral vasculature.

A rise in [Ca^2+^]_i_ can elicit apoptosis through elevating mitochondrial Ca^2+^ content and depolarizing ΔΨ_m_, resulting in the release of cytochrome C and activation of caspases [13]. In the present experiments, the extent of cell death was related to the progressive rise in [Ca^2+^]_i_ during H_2_O_2_ exposure. That this [Ca^2+^]_i_ response was reduced in WD vs. CD mice (Figure 4) but not in DIO vs. SD mice, again points to a role for processed carbohydrates in vascular adaptation to H_2_O_2_ exposure.

### 4.2. Changes in TRPV4 and TRPC3 Channel-Mediated Ca^2+^ Entry Contribute to Differences in H_2_O_2_-Induced Cell Death

The integral role of TRPV4 and TRPC3 channels as routes of Ca^2+^ influx leading to apoptosis in the vascular wall during H_2_O_2_ exposure [14] is confirmed by the present experiments. Inhibition of either TRPV4 (Figure 5) or TRPC3 (Figure 6) channels limited SMC and EC death in CD, but not WD mice that had adapted by reducing Ca^2+^ influx. Finding that either TRP channel inhibitor was able to prevent cell death suggests an interaction between the respective channel subunits. Functional TRP channels are composed of tetramers and both TRPC3 and TRPV4 subunits are capable of forming functional heteromeric channels [41]. While a distinct heteromer remains to be identified in the context of this study, both TRPC3 and TRPV4 form functional tetramers with TRPC1 [42,43]. A key question for future studies is whether (and if so, how) TRP channel expression is affected by WD in vascular cells.

Src family kinases can be activated by oxidative stress [32,44] which has been linked to apoptosis in epithelial cells [45]. These kinases can activate both TRPV4 through phosphorylation of Tyr^110^ [32] and TRPC3 channels though phosphorylation of Tyr^226^ [46,47]. Unlike the protective effect of TRP channel inhibition on both cell layers of PCAs (Figure 5 and Figure 6), Src kinase inhibition reduced cell death in SMCs, but not ECs (Figure 7). Given the role of TRPV4 and TRPC3 channels in mediating death of respective cell types during H_2_O_2_ exposure, this differential outcome suggests that alternative mechanisms of channel activation are involved in ECs vs. SMCs. Although H_2_O_2_ can activate different TRP channel isoforms through the oxidation of cysteine residues [48,49], it remains to be determined whether such activation occurs in ECs. Other signaling events may include the activation of Src kinase by H_2_O_2_ or oxidation of Ca^2+^/calmodulin-dependent protein kinase, which can be stimulated by oxidative stress and thereby activate Src kinase [50,51]. Nevertheless, our finding that Src kinase inhibition limited SMC death and attenuated the rise in [Ca^2+^]_i_ supports a role for Src kinase activity in transducing the signal from H_2_O_2_ to SMCs of mouse PCAs.

The mechanism(s) by which WD and chronic oxidative stress reduce Ca^2+^ influx and thereby enhance resilience to H_2_O_2_ remain(s) to be fully defined. Obesity has been linked to reduced TRPV4-dependent dilation in mesenteric arteries resulting from peroxynitrite-dependent inactivation of AKAP_150_ [52]. It is also possible that adaptations of the plasma membrane facilitate the greater cell survival in PCAs from WD mice vs. CD mice. Both TRP channels [53] and Src kinases [54] can be regulated by local lipid domains. Recent findings show that oxidized phospholipids increase stress tolerance in ECs, thereby limiting cell death [55]. Whether the protective effect of WD on Ca^2+^ influx and vascular cell death can be explained by changes in membrane lipids that affect TRP channels and/or Src kinases during acute oxidative stress remains to be addressed.

## 5. Conclusions

In cerebral arteries from adult mice, different high fat diets dissimilarly alter the resilience to acute oxidative stress. Whereas a high-fat-diet-induced obesity model did not affect susceptibility to H_2_O_2_, western-style diet, which is high in processed carbohydrates as well as fat, increased vascular ROS production and protected SMCs and ECs during acute H_2_O_2_ exposure. This enhanced vascular resilience to oxidative stress is mediated by limiting Ca^2+^ entry through TRP channels, with Src kinase having an integral role in SMCs. Although the incidence of ischemic stroke has been reported to be lower in men consuming a high fat diet [56], not all studies agree [57]. The present data suggest that such inconsistencies between studies may reflect the influence of processed carbohydrates in addition to elevated fat consumption.

## Figures and Tables

**Figure 1 antioxidants-12-01433-f001:**
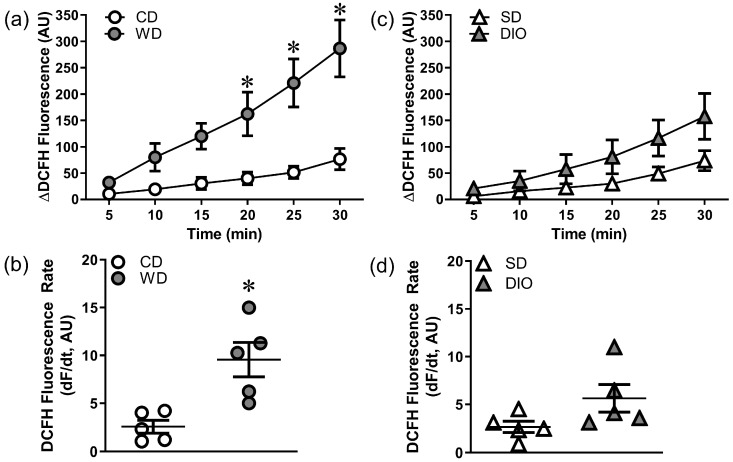
High fat diets augment ROS production in PCAs. Values represent DCFH fluorescence over 30 min H_2_O_2_ exposure. (**a**) Summary data for changes in ROS production in PCAs from CD and WD mice. (**b**) Rate of DCFH fluorescence accumulation [dF/d*t*, where F is fluorescence and *t* is time (min)] for vessels in (**a**). WD increases ROS production vs. CD. (**c**) Summary data for changes in ROS production in PCAs from SD and DIO mice. (**d**) Rate of DCFH fluorescence accumulation for vessels in (**c**); there is a trend (*p* = 0.06) for DIO to increase ROS production vs. SD. Summary values are means ± SE; *n* = 5 vessels (each from a different mouse)/group. * *p* < 0.05 vs. CD.

**Figure 2 antioxidants-12-01433-f002:**
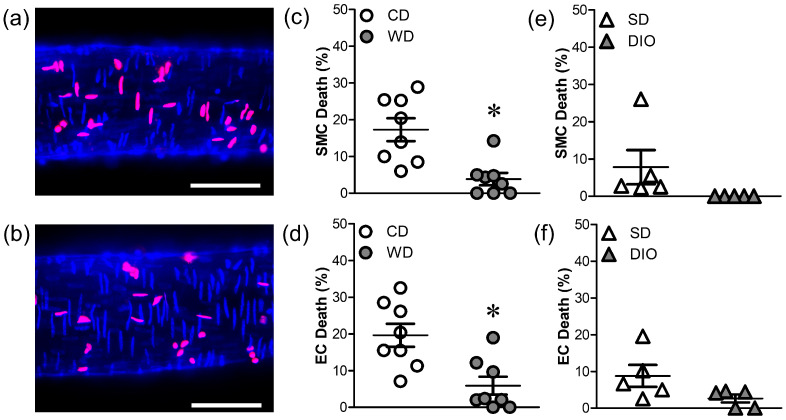
WD protects against H_2_O_2_-induced vascular cell death. Merged image of Hoechst 33,342 (blue) and propidium iodide (red) staining of cell nuclei in PCAs from a CD (**a**) and WD (**b**) mouse after 50 min H_2_O_2_ exposure. Scale bars = 50 µm. (**c**–**f**) Percentage of dead SMCs (**c**,**e**) and ECs (**d**,**f**) in PCAs from WD (**c**,**d**) and DIO (**e**,**f**) mice and respective controls following H_2_O_2_ exposure. WD significantly attenuated SMC and EC death while DIO did not. Summary values are means ± SE; *n* = 5–8 vessels/group. * *p* < 0.05 vs. CD.

**Figure 3 antioxidants-12-01433-f003:**
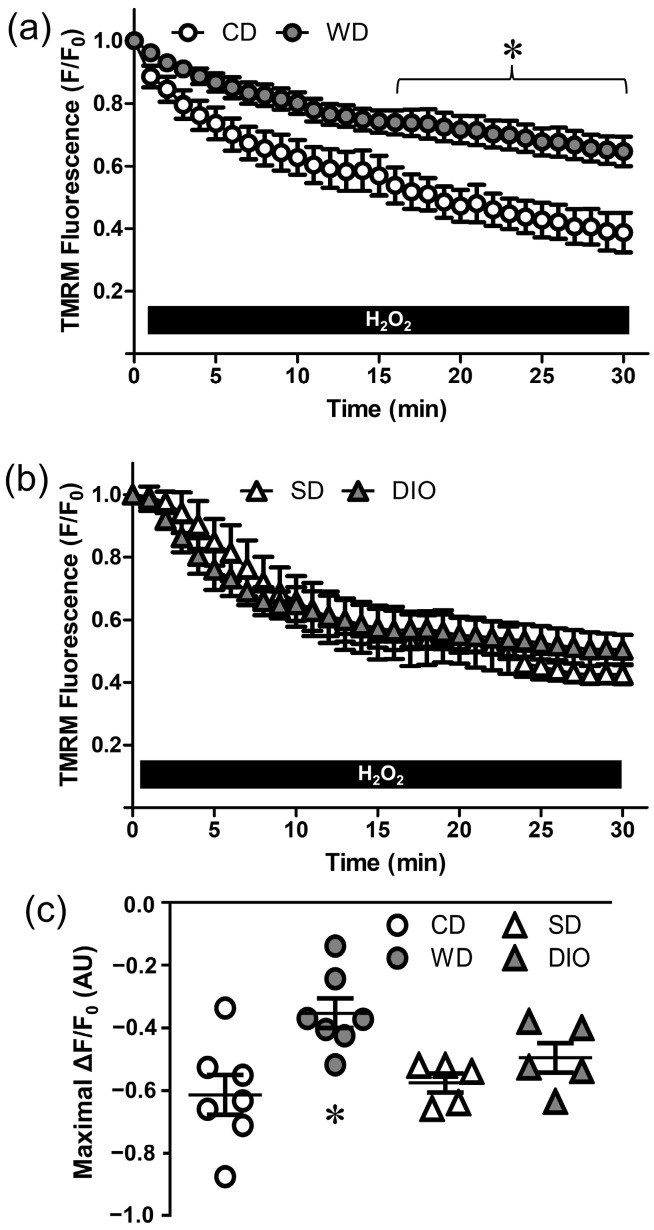
Western diet attenuates mitochondrial depolarization during H_2_O_2_ exposure. Changes in mitochondrial membrane potential (ΔΨ_m_) during exposure to H_2_O_2_ in PCAs from WD and CD mice (**a**), and from DIO and SD mice (**b**). The decline in TMRM fluorescence corresponds to depolarization (loss) of ΔΨ_m_. (**c**) Maximal ΔΨ_m_ depolarization following 30 min H_2_O_2_ exposure (ΔF/F_0_) in PCAs from each group. WD reduced ΔΨ_m_ depolarization but DIO did not. Summary values are means ± SE; *n* = 5–7 vessels/group. * *p* < 0.05 vs. CD.

**Figure 4 antioxidants-12-01433-f004:**
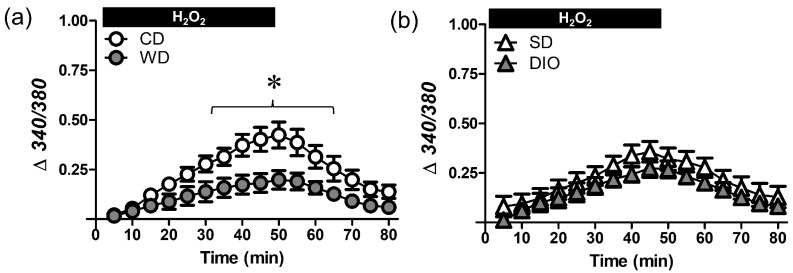
Effect of H_2_O_2_ exposure on vessel wall [Ca^2+^]_i_. Fura 2 fluorescence (Δ340/380) during 50 min H_2_O_2_ exposure (200 µM) followed by 30 min in standard PSS. Data are for SMCs in PCAs from WD and CD (**a**) and DIO and SD (**b**) mice. Summary values are means ± SE; *n* = 5 vessels/group. * *p* < 0.05 vs. CD.

**Figure 5 antioxidants-12-01433-f005:**
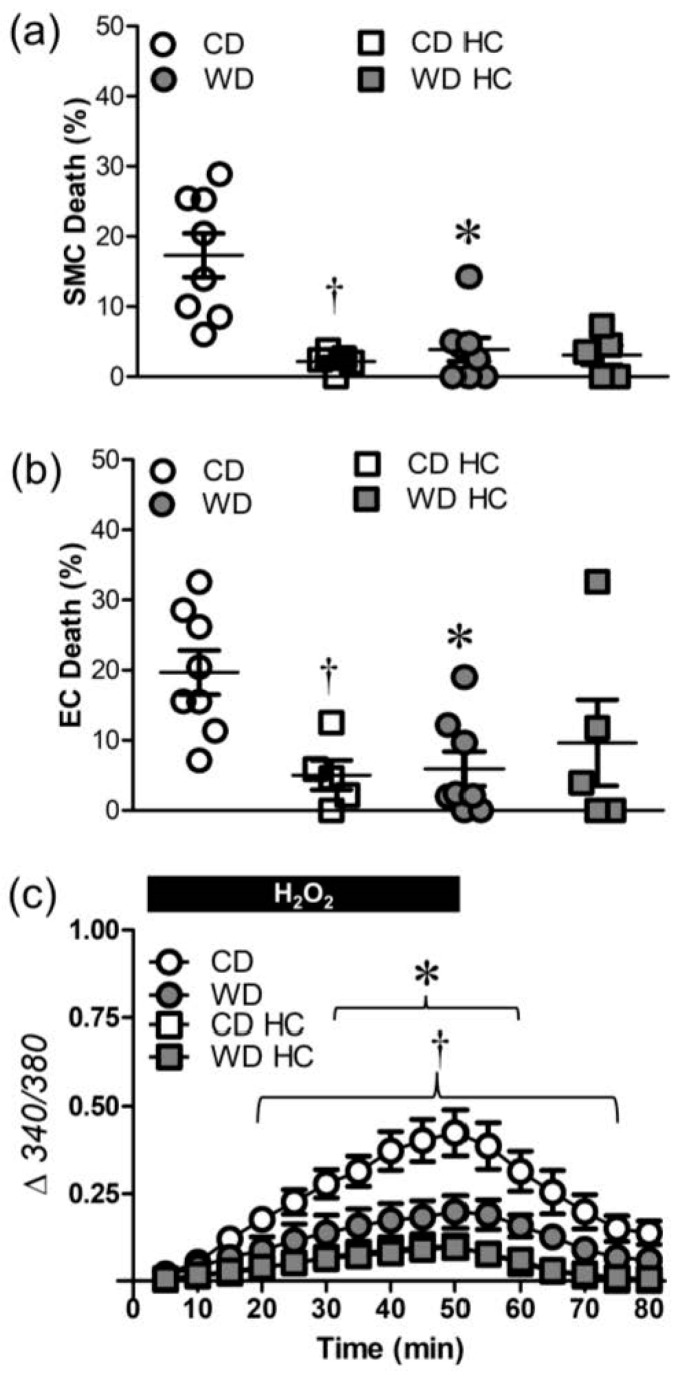
TRPV4 channel inhibition attenuates cell death and H_2_O_2_-induced Ca^2+^ entry. (**a**) SMC and (**b**) EC death following H_2_O_2_ exposure in PCAs from WD and CD mice in the presence of the TRPV4 inhibitor HC-067047 (HC, 1 µM) or its vehicle control. (**c**) Fura 2 fluorescence (Δ340/380) in in SMCs of PCAs the absence and presence of HC (*Note: CD HC data obscured by WD*). Summary values are means ± SE; *n* = 5–8 vessels/group. * *p* < 0.05 WD vs. CD. ^†^ *p* < 0.05 CD HC vs. CD.

**Figure 6 antioxidants-12-01433-f006:**
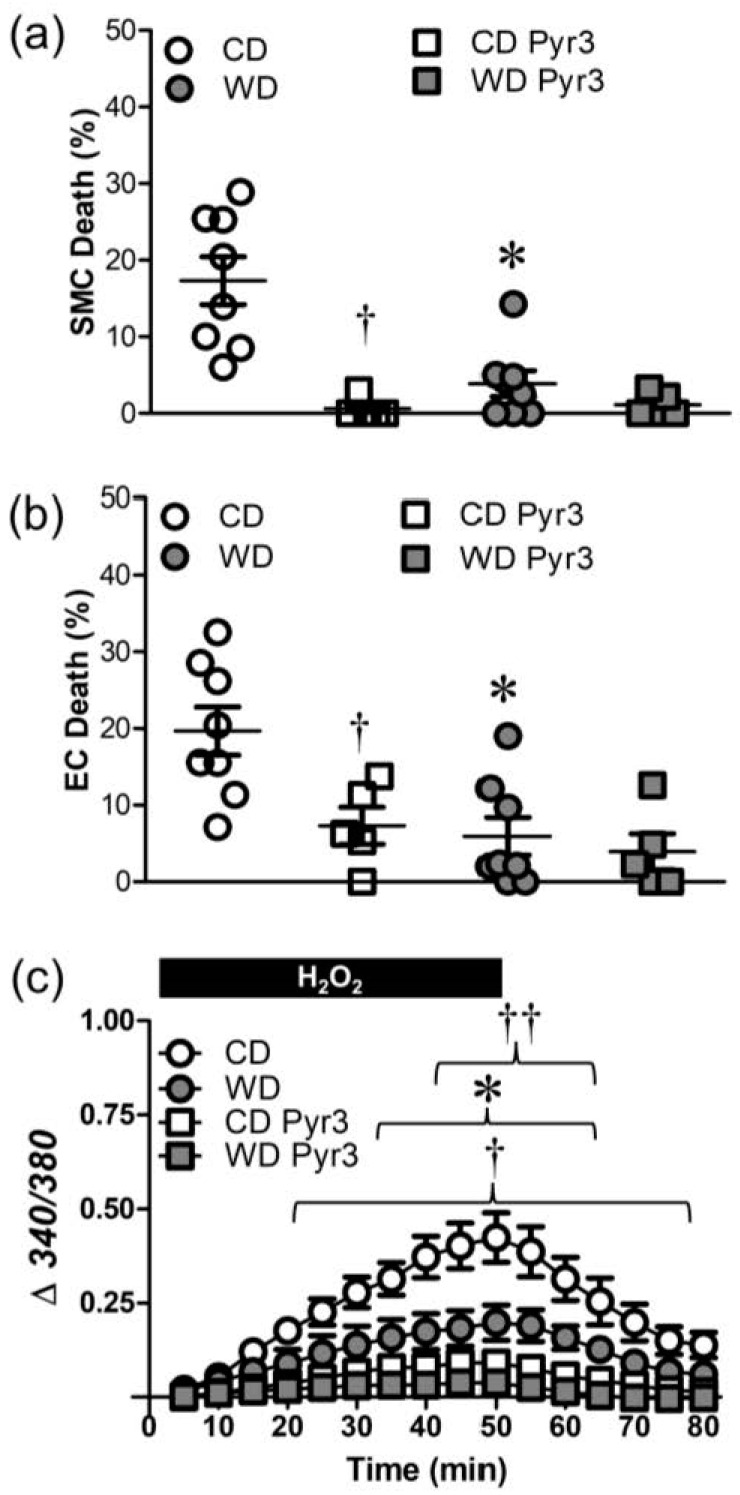
TRPC3 channel inhibition attenuates cell death and H_2_O_2_-induced Ca^2+^ entry. (**a**) SMC and (**b**) EC death following H_2_O_2_ exposure in PCAs from WD and CD mice in the presence of the TRPC3 inhibitor Pyr3 (1 µM) or its vehicle control. (**c**) Fura 2 fluorescence (Δ340/380) in SMCs of PCAs in the absence and presence of Pyr3. Summary values are means ± SE; *n* = 5–8 vessels/group. * *p* < 0.05 WD vs. CD. ^†^ *p* < 0.05 CD Pyr3 vs. CD. ^††^ *p* < 0.05 WD Pyr3 vs. WD.

**Figure 7 antioxidants-12-01433-f007:**
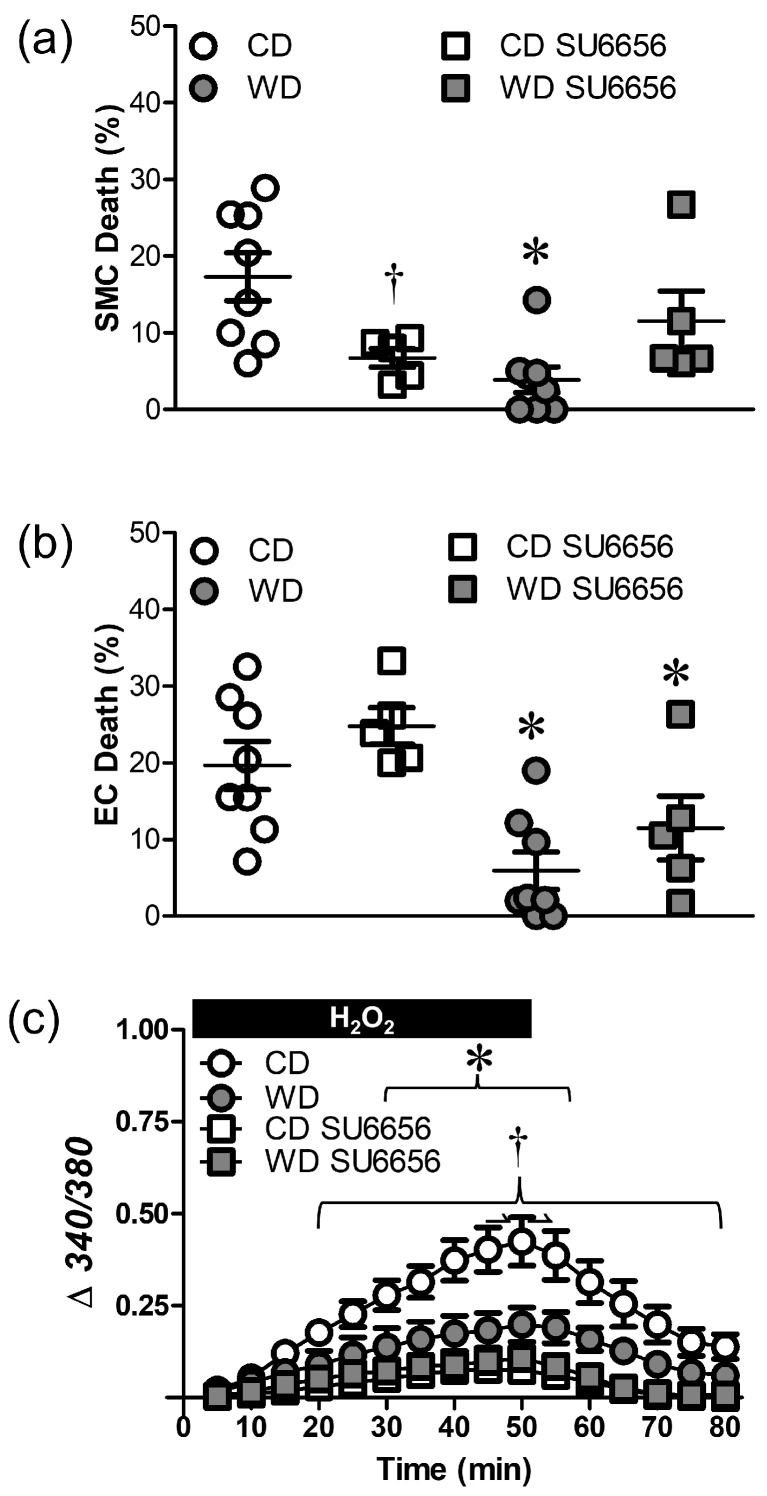
Src family kinases contribute to H_2_O_2_-induced cell death and Ca^2+^ entry. (**a**) SMC and (**b**) EC death to H_2_O_2_ in PCAs from WD and CD mice in the presence of the Src kinase inhibitor SU6656 (10 µM) or its vehicle control. (**c**) Fura-2 fluorescence (Δ340/380) in SMCs of PCAs in the absence or presence of SU6656 during H_2_O_2_ exposure. Summary values are means ± SE; *n* = 5–8 vessels/group. * *p* < 0.05, WD vs. CD. ^†^ *p* < 0.05, CD SU6656 vs. CD.

**Table 1 antioxidants-12-01433-t001:** WD and DIO increase in body weight (BW) on the day of experiments. Data are means ± SE. * *p* < 0.05 WD vs. CD. ^#^ *p* < 0.05 DIO vs. SD.

Group	BW (g)	*n*
CD	29.9 ± 0.3	15
WD	45.2 ± 0.9 *	15
SD	33.6 ± 1.1	5
DIO	45.1 ± 1.6 ^#^	5

## Data Availability

Data supporting the findings of this study are located at https://doi.org/10.7910/DVN/RRROZ1 (accessed on 10 May 2023).

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
