# Peer review of "Differential Effects of High Fat Diets on Resilience to H2O2-Induced Cell Death in Mouse Cerebral Arteries: Role for Processed Carbohydrates"

_antioxidants, 2023, doi:10.3390/antiox12071433_

Round 1
Reviewer 1 Report
General comments
It is a quiet interesting study. I believe that this manuscript tries to investigate if different types of high fat diets could affect H2O2-induced cell death in mouse model. The results in the manuscript would help to understand the effects of WD and DIO in mice model. The strength of the study is the wealth of experimental results. However, there are also some of weaknesses. The following is my comments and critique.
Major
1. For mice study, how many times repeated? It looks just once. If so, please repeat at least twice.
2. It is fine to understand there are some different effects between WD and DIO in mice model. I think the author should explain and describe more detailed possible pathways or targets why or how WD works, not DIO in Discussion part.
3. Have you checked antioxidant related gene expression (protein or mRNA) with WD or DIO?
Minors
1. For table 1, better to show body weight changes per week in Figure. And also more detailed information in legend.
2. In Figure 2C, a star symbol (*) location should be fixed.
3. For statistics symbol (*, †) in Figure 5, 6 and 7, please make a clear statement.
For example, † p<0.05 HC vs CD. HC means which HC?, CD-HC? or WD-HC?
should use English editing service
Author Response
The authors thank the Managing Editor and our reviewers for comments that have helped us to strengthen the presentation of our research findings. Each comment is copied below followed by the authors’ response. Revisions to the manuscript are identified by line number and red font with reference to the red-lined version of this revised submission. Responses for all reviewers can be seen in the attachment.
Reviewer 1
Major
- For mice study, how many times repeated? It looks just once. If so, please repeat at least twice.
Response: All experimental protocols were repeated in vessels from n=5-8 mice. As now clarified, each vessel studied in a given protocol is from a different mouse (page 4; line 189 and page 5; line 214). Contralateral arteries from the same mouse were used for separate procedures. The specific number of observations (individual vessels) for each figure is included in the figure captions.
- It is fine to understand there are some different effects between WD and DIO in mice model. I think the author should explain and describe more detailed possible pathways or targets why or how WD works, not DIO in Discussion part.
Response: We have expanded upon how WD differs from the DIO model. Potential differences in antioxidant response (pages 9-10; lines 339-349) and modifications to mitochondria (page 10; lines 359-366) are now addressed in the revised Discussion.
Have you checked antioxidant related gene expression (protein or mRNA) with WD or DIO?
Response: We have yet to perform experiments evaluating antioxidant related gene expression in cerebral arteries from WD and DIO mice. In response to this question, the discussion now includes additional information, particularly the potential role for Nrf2 (Strom et al., 2016; Ngo & Duennwald, 2022). The importance of studying the antioxidant response is now also addressed (page 9; lines 342-342).
Minor
- For table 1, better to show body weight changes per week in Figure. And also more detailed information in legend?
Response: Body weight was measured on the day of each experiment, so we do not have data for changes in body weight over time. For table 1, we have clarified this in the title and provided more detailed descriptions of statistical comparisons.
Figure 2C, a star symbol (*) location should be fixed?
Response: Thanks for this catch, which appears to apply to Figure 3C and is fixed.
- For statistics symbol (*, †) in Figure 5, 6 and 7, please make a clear statement. For example, † p<0.05 HC vs CD. HC means which HC?, CD-HC? or WD-HC?
Response: The comparisons in the legends of Figures 5-7 are revised to clarify the diet and treatment for respective groups.
References:
Ngo V & Duennwald ML. (2022). Nrf2 and oxidative stress: A general overview of mechanisms and implications in human disease. Antioxidants 11, 2345.
Strom J, Xu B, Tian X & Chen MQ. (2016). Nrf2 protects mitochondrial decay by oxidative stress. FASEB J 30, 66-80.

Reviewer 2 Report
The authors have suggested "differential effects of high fat diets on resilience to H2O2-induced cell death in mouse cerebral arteries" by clarifying the role for processed carbohydrates. They presented ECs and SMCs of posterior cerebral arteries (PCAs) from mice consuming a diet high in fat and processed carbohydrate would affect resilience to acute oxidative stress. The present data shows that the processed carbohydrate in addition to elevated fat consumption influence the H2O2-induced cell death.
Would you make it clear what groups you compare in Figs 6 and 7?
†P<0.05 Pyr3 vs. CD?
†P<0.05, SU6656 vs. CD?
Author Response
The authors thank the Managing Editor and our reviewers for comments that have helped us to strengthen the presentation of our research findings. Each comment is copied below followed by the authors’ response. Revisions to the manuscript are identified by line number and red font with reference to the red-lined version of this revised submission. Responses to all reviewers can be seen in the attachment.
Reviewer 2
- Would you make it clear what groups you compare in Figs 6 and 7?
†P<0.05 Pyr3 vs. CD?
†P<0.05, SU6656 vs. CD?
Response: The legends in Figures 5-7 are revised to clarify comparisons of the diet and treatment for respective groups.

Reviewer 3 Report
The work by Norton et al. describes how smooth muscle and endothelial cells in mice adapt during western style high-fat diet (WD) exposure to become more resilient to acute oxidative stress. The authors suggest that WD consumption leads to adaptations that limit calcium influx and vascular cell death during exposure to acute oxidative bouts. Although the topic is interesting, several methodological issues should be addressed to solidify this statement, especially related to the oxidative stress response.
Major points:
1. How do you know that there was oxidative stress in your model? Have you done any experiments for biomolecule oxidation? Lipid peroxidation (4-HNE), protein carbonylation, GSH, and GSSG levels? Even if using 200 micromolar of H2O2, the authors still need to provide results that demonstrate oxidative stress. Maybe, the results observed are caused by H2O2induced modulating Ca2+-related pathways, as CAMK is modulated by ROS.
2. Materials and Methods: Topic 2.3 – and results in Figure 2: Before checking cell death, have you performed cell viability assays before treating the cells with H2O2? In the manuscript, it was not presented, so it is hard to understand if the lower cell death by H2O2 is due to a lower cell viability in the basal state.
3. Materials and Method: Topic 2.6: Why haven’t you isolated mitochondria? How could you interpret that the TMRM was targeting only mitochondria? Could you present these data in isolated mitochondria?
4. ROS production: Is ROS production is increased via mitochondria, NADPH Oxidase, or other sources? I recommend you isolate mitochondria and perform membrane potential analysis and ROS production. To address H2O2 production, please address NOX activity, since it is one of the main sources of H2O2. Another complementary experiment is measuring the activity of superoxide dismutase activity.
5. Results – Figure 2: “WD protects against H2O2-induced vascular cell death” – Does WD protect, or has the higher ROS production induced by WD already created a pro-oxidative environment (ad it was shown in Figure 1) and, consequently activated some compensatory mechanisms of cell protection, for instance, antioxidant capacity via Nrf2? I strongly recommend you present the results from H2O2 non-treated cells to check for vascular cell death and explore the mechanisms related to this possible “protective effect” (i.e. antioxidant capacity analysis).
Minor points:
6. Results – Figure 3 – Lines 232-233: Does lower depolarization capacity indicate a protective or deleterious effect? In general, reduced depolarization capacity is associated with lower responsiveness and functional capacity. Also, as mentioned in the manuscript (line 230: “Depolarization is a key signaling event mediating cell death”, but what is the relationship? In general, lower depolarization is associated with higher mitochondrial capacity. Please proofread and review this statement, your data is providing strong pieces of evidence that WD is not protective.
7. Introduction – Please, provide a better explanation of the general and specific aspects and knowledge that will support your findings. I strongly recommend you rewrite the Introduction. Could you characterize oxidative stress and explain how it can lead the cells to death? Which are the main mechanisms involved? Why oxidative stress is important to understand your intervention? How is oxidative stress connected with the apoptotic response? The introduction is not well-written, too short, and the sentences/ideas are not well linked.
8. Abstract – There are too many abbreviations, and the main findings were not well-presented. In line 17, add the “)” in the standard abbreviation “(SD”.
9. Materials and Methods – Animal care and use: Have you measured the food intake?
10. Results – Line 185: Take care with some adjectives, I would recommend you remove the word “strong”. There is a trend.
11. Results – Please, move Figure 1 to topic 3.1. It was confusing to understand that Figure 1 was related to topic 3.1 as soon as it is in topic 3.2.
Author Response
The authors thank the Managing Editor and our reviewers for comments that have helped us to strengthen the presentation of our research findings. Each comment is copied below followed by the authors’ response. Revisions to the manuscript are identified by line number and red font with reference to the red-lined version of this revised submission. Individual responses for all review comments can be found in the attachment.
Reviewer 3
Major points:
- How do you know that there was oxidative stress in your model? Have you done any experiments for biomolecule oxidation? Lipid peroxidation (4-HNE), protein carbonylation, GSH, and GSSG levels? Even if using 200 micromolar of H2O2, the authors still need to provide results that demonstrate oxidative stress. Maybe, the results observed are caused by H2O2-induced modulating Ca2+-related pathways, as CAMK is modulated by ROS.
Response: Research with the WD model has shown increased oxidative stress assessed by 3-nitrotyrosine staining in the aorta (DeMarco et al., 2015; Manrique et al., 2016). The authors appreciate the identification of additional markers of oxidation. Although these have yet to be evaluated in our system, the role of CAMK as an activator of Src kinases through H2O2 exposure (Wang et al., 2003; Anderon, 2015) is addressed in the revised discussion (page 10; 396-398).
- Materials and Methods: Topic 2.3 – and results in Figure 2: Before checking cell death, have you performed cell viability assays before treating the cells with H2O2? In the manuscript, it was not presented, so it is hard to understand if the lower cell death by H2O2 is due to a lower cell viability in the basal state.
Response: Yes, these controls are essential and this is an important point to clarify. Our time controls for mouse PCAs have verified that PCAs studied under these conditions exhibit <1% cell death after 50 min in the absence of H2O2; this information is now included in the methods to confirm that PCAs were not damaged as a result of dissection and cannulation (page 3; lines 122-124). A sentence is also added to the results to clarify that initial viability (prior to treating with H2O2) was similar (1-3% cell death) for all treatment groups (page 5; line 216).
- Materials and Method: Topic 2.6: Why haven’t you isolated mitochondria? How could you interpret that the TMRM was targeting only mitochondria? Could you present these data in isolated mitochondria?
Response: In light of the present findings, we too are intrigued by the potential for mitochondrial adaptations and how they may relate to cell death. New experiments are being developed to evaluate how mitochondria respond to oxidative stress; studying isolated mitochondria will be a focus of this new research direction.
Our control experiments pertaining to the localization of TMRM to the mitochondria in mouse cerebral arteries are presented in another manuscript which is also under review, where we have evaluated how advanced age and sex affect cell death in PCAs exposed to H2O2. In those experiments we show that TMRM fluorescence overlaps that that of mitotracker, which is consistent with images from rat pulmonary arteries and human pulmonary arterial smooth muscle cells (Snow et al., 2020). We now include data from experiments showing that the protonophore FCCP depolarizes ΔΨm in PCAs from control (CD) mice (pages 3-4, lines 147-149; page 6; lines 235-239). These data support TMRM serving as an effective index of ΔΨm in the present study.
- ROS production: Is ROS production is increased via mitochondria, NADPH Oxidase, or other sources? I recommend you isolate mitochondria and perform membrane potential analysis and ROS production. To address H2O2 production, please address NOX activity, since it is one of the main sources of H2O2. Another complementary experiment is measuring the activity of superoxide dismutase activity.
Response: A key finding of the present study is that chronic vascular oxidative stress leads to resilience of cerebral vascular smooth muscle and endothelial cells acutely exposed to H2O2. The authors agree that the source(s) of increased ROS production (i.e. mitochondrial, NADPH oxidase, reduced scavenging) need to be evaluated. This is a central question for new experiments as now identified in the discussion (page 9, lines 344-346) and these recommendations are appreciated.
- Results – Figure 2: “WD protects against H2O2-induced vascular cell death” – Does WD protect, or has the higher ROS production induced by WD already created a pro-oxidative environment (ad it was shown in Figure 1) and, consequently activated some compensatory mechanisms of cell protection, for instance, antioxidant capacity via Nrf2? I strongly recommend you present the results from H2O2 non-treated cells to check for vascular cell death and explore the mechanisms related to this possible “protective effect” (i.e. antioxidant capacity analysis).
Response: Statements in the revised results (page 5; line 216) confirm that in the absence of H2O2, cell death in our preparations is minimal regardless of the diet consumed. In our experimental design, where the vessel is continuously supplied with fresh H2O2 (200 µM), in the superfusion solution, we think it is unlikely that antioxidant capacity alone is sufficient to alter cell death. The revised Discussion (pages 9-10; lines 339-349) addresses this point, and highlights the potential for antioxidant response elements such as Nrf2 to be mediators of protection in this setting (Strom et al., 2016; Ngo & Duennwald, 2022).
Minor points:
- Results – Figure 3 – Lines 232-233: Does lower depolarization capacity indicate a protective or deleterious effect? In general, reduced depolarization capacity is associated with lower responsiveness and functional capacity. Also, as mentioned in the manuscript (line 230: “Depolarization is a key signaling event mediating cell death”, but what is the relationship? In general, lower depolarization is associated with higher mitochondrial capacity. Please proofread and review this statement, your data is providing strong pieces of evidence that WD is not protective.
Response: We agree that this point required clarification. While western diet may lead to disruption of mitochondrial function/reduced functional capacity in the absence of acute oxidative stress (Jaiswal et al., 2015), such changes may attenuate the ability of mitochondria to depolarize in response to oxidative stress and thereby enhance cellular resilience to a pathophysiological response by preserving Ψm. Alternatively it has been shown that nutrient excess can enhance ΔΨm (Liesa & Shirihai, 2013) which may also limit depolarization to H2O2. The revised discussion now addresses these relationships (page 10; lines 359-366).
- Introduction – Please, provide a better explanation of the general and specific aspects and knowledge that will support your findings. I strongly recommend you rewrite the Introduction. Could you characterize oxidative stress and explain how it can lead the cells to death? Which are the main mechanisms involved? Why oxidative stress is important to understand your intervention? How is oxidative stress connected with the apoptotic response? The introduction is not well-written, too short, and the sentences/ideas are not well linked.
Response: This feedback is sincerely appreciated. The revised introduction is expanded to improve the flow and clarity of ideas; also, to elaborate key points including the characterization of oxidative stress, describing why oxidative stress is integral to our intervention, and addressing how oxidative stress elicits apoptosis (pages 1-2; lines 32-64).
- Abstract – There are too many abbreviations, and the main findings were not well-presented. In line 17, add the “)” in the standard abbreviation “(SD”.
Response: We have revised the abstract (page 1; lines 13-27) to improve presentation of the main findings and reduced the number of abbreviations. We have additionally inserted the missing “)”.
- Materials and Methods – Animal care and use: Have you measured the food intake?
Response: No, we did not measure food intake in the mice studied here. The DIO and SD mice were purchased from (and fed at) Jackson Laboratories.
- Results – Line 185: Take care with some adjectives, I would recommend you remove the word “strong”. There is a trend.
Response: The word “strong” was removed from the statement (page 5, line 203).
- Results – Please, move Figure 1 to topic 3.1. It was confusing to understand that Figure 1 was related to topic 3.1 as soon as it is in topic 3.2.
Response: Agreed. Figure 1 is now in the 3.1 topic section to avoid confusion.
References
Anderon ME. (2015). Oxidant stress promotes disease by activating CaMKII. J Mol Cell Cardiol 89, 160-167.
DeMarco VG, Habibi J, Jia G, Aroor AR, Ramirez F, Martinez-Lemus LA, Bender SB, Garro M, Hayden MR, Sun Z, Meininger GA, Manrique C, Whaley-Connell & Sowers JR. (2015). Low dose mineralocorticoid receptor blockade prevents western diet-induced arterial stiffening in mice. Hypertension 66, 99-107.
Jaiswal N, Maurya CK, Arha D, Avisetti DR, Parathapan A, Raj PS, Raghu KG, Kalivendi SV & Tammrakar AK. (2015). Fructose induces mitochondrial dysfunction and triggers apoptosis in skeletal muscle cells by provoking oxidative stress. Apoptosis 20, 930-947.
Liesa M & Shirihai OS. (2013). Mitochondrial dynamics in the regulation of nutrient utilization and energy expenditure. Cell Metabolism 17, 491-506.
Manrique C, Habibi J, Aroor AR, Jia G, Hayden MR, Garro M, Martinez-Lemus LA, Ramirez-Perez FI, Klein T, Meininger GA & DeMarco VG. (2016). Dipeptidyl peptidase-4 inhibition with linagliptin prevents western diet-induced vascular abnormalities in female mice. Cardiovasc Diabetol 15, 94.
Ngo V & Duennwald ML. (2022). Nrf2 and oxidative stress: A general overview of mechanisms and implications in human disease. Antioxidants 11, 2345.
Snow JB, Norton CE, Sands MA, Weise Cross L, Yan S, Herbert LM, Sheak JR, Gonzales Bosc LV, Walker BR, Kanagy NL, Jernigan NL & Resta TC. (2020). Intermittent hypoxia aurments pulmonary vasoconstrictor reactivity through PCKB/mitochondrial oxidant signaling. Am J Respir Cell Mol Biol 62, 732-746.
Strom J, Xu B, Tian X & Chen MQ. (2016). Nrf2 protects mitochondrial decay by oxidative stress. FASEB J 30, 66-80.
Wang Y, Mishra R & Simonson MS. (2003). Ca2+/calmodulin-dependent protein kinase II stimulates c-fos transcription and DNA synthesis by a Src-based mechanism in glomerular mesangial cells. J Am Soc Nephrol 14, 28-36.

Round 2
Reviewer 1 Report
Thanks for all author's efforts to revise the MS.
Reviewer 3 Report
Thank you for responding to my comments.